

# Urban aerosol chemistry at a land-water transition site during summer – Part 2: Aerosol pH and liquid water content

Michael A. Battaglia, Jr.[1,a], Nicholas Balasus[1], Katherine Ball[1], Vanessa Caicedo[2], Ruben Delgado[2], Annmarie G. Carlton[3] , and Christopher J. Hennigan[1*]

[1]Department of Chemical, Biochemical, and Environmental Engineering, University of Maryland, Baltimore County

[2]Joint Center for Earth Systems Technology, University of Maryland, Baltimore County

[3]Department of Chemistry, University of California, Irvine

[a]Current affiliation: School of Earth and Atmospheric Sciences, Georgia Institute of Technology

*Correspondence*: C. J. Hennigan (hennigan@umbc.edu)

**Abstract**
3          Particle acidity (aerosol pH) is an important driver of atmospheric chemical processes
and the resulting effects on human and environmental health.  Understanding the factors that
control aerosol pH is critical when enacting control strategies targeting specific outcomes.  This
study characterizes aerosol pH at a land-water transition site near Baltimore, MD during summer
2018 as part of the second Ozone Water-Land Environmental Transition Study (OWLETS-2)
field campaign.  Inorganic fine mode aerosol composition, gas-phase $NH_3$ measurements, and all
relevant meteorological parameters were used to characterize the effects of temperature, aerosol
liquid water (ALW), and composition on predictions of aerosol pH.  Temperature, the factor
linked to the control of $NH_3$ partitioning, was found to have the most significant effect on aerosol
pH during OWLETS-2.  Overall, pH varied with temperature at a rate of -0.047 K$^{-1}$ across all
observations, though the sensitivity was -0.085 K$^{-1}$ for temperatures > 293 K.  ALW had a minor



effect on pH, except at the lowest ALW levels (< 1 μg m$^{-3}$) which caused a significant increase
in aerosol acidity (decrease in pH).  Aerosol pH was generally insensitive to composition ($SO_4^{2-}$,
$SO_4^{2-}$:$NH_4^+$ , Tot-$NH_3$ = $NH_3$ + $NH_4^+$), consistent with recent studies in other locations.  In a
companion paper, the sources of episodic $NH_3$ events (95[th] percentile concentrations, $NH_3$ > 7.96
μg m$^{-3}$) during the study are analyzed; aerosol pH was higher by only ~0.1-0.2 pH units during
these events compared to the study mean.  A case study was analyzed to characterize the
response of aerosol pH to nonvolatile cations (NVCs) during a period strongly influenced by
primary Chesapeake Bay emissions.  Depending on the method used, aerosol pH was estimated
to be either weakly (~0.1 pH unit change based on $NH_3$ partitioning calculation) or strongly
(~1.4 pH unit change based on ISORROPIA thermodynamic model predictions) affected by
NVCs.  The case study suggests a strong pH gradient with size during the event and underscores
the need to evaluate assumptions of aerosol mixing state applied to pH calculations.  Unique
features of this study, including the urban land-water transition site and the strong influence of
$NH_3$ emissions from both agricultural and industrial sources, add to the understanding of aerosol
pH and its controlling factors in diverse environments.



## 1 Introduction

The acidity, or pH, of atmospheric aerosols affects the chemical and physical properties
of airborne particles, and thus, their impacts on climate and health (Pye et al., 2020).  The gas-
particle partitioning of semi-volatile acidic and basic compounds – notably $NH_3$, $HNO_3$, HCl,
and organic acids – depends in part on aerosol pH, which directly affects the particulate matter
(PM) mass concentration (Nenes et al., 2020).  The solubility of many particulate components is
pH-dependent, including metals and nutrients, with implications for particle toxicity and nutrient
deposition to ecosystems (Fang et al., 2017; Kanakidou et al., 2016).  The optical properties of
light-absorbing organic compounds, known as brown carbon, can exhibit a strong pH-
dependence, which directly affects their climate impacts (Phillips et al., 2017).  Given the
importance of aerosol pH for atmospheric processes and the limitation in estimating acidity with
proxies (e.g., ion balances), there has been increased effort in recent years to identify the factors
that affect pH and to characterize temporal and spatial variations in the atmosphere (Hennigan et
al., 2015).
Globally, aerosol pH is often quite acidic due to the ubiquity and abundance of strong
acids like $H_2SO_4$, $HNO_3$, and HCl (Pye et al., 2020).  Ammonia ($NH_3$) is unique among species
that affect pH because it is the most abundant basic compound in the atmosphere.  $NH_3$
partitioning is controlled by the concentration of strong acids and by the ambient temperature
and relative humidity, hence, the dependence of pH on both composition and meteorology
(Zheng et al., 2020).  This explains why $NH_3$ can exist partially in the gas phase even when the
aerosol pH is highly acidic (Weber et al., 2016).  Due to its abundance and semi-volatile
properties, $NH_3$ was identified as the most important buffering agent in aerosols across locations
with diverse emissions, composition, and climatology (Zheng et al., 2020).  In single-phase



aqueous particles, organic compounds have a minor effect on pH (Battaglia Jr et al., 2019),
though this is not the case for particles that have undergone liquid-liquid phase separation (Pye et
al., 2018). Non-volatile cations (NVC) typically contribute a minor fraction of PM mass but can
be critical for accurate predictions of pH, especially if NVC concentrations are overestimated
(Vasilakos et al., 2018). Globally, NVC are most important in regions heavily impacted by dust
emissions (Pye et al., 2020), but have minor effects on pH in other regions (Tao and Murphy,
2019; Zheng et al., 2020).

Aerosol pH is also strongly affected by meteorological factors. Equilibrium constants,

including those that determine the gas-particle partitioning and aqueous dissociation of semi-
volatile acids and bases, are temperature dependent. Temperature is a dominant factor driving
variability in the seasonal and diurnal cycling of pH (Guo et al., 2015; Tao and Murphy, 2019).
Temperature gradients, as occur in and around urban areas, can also drive large differences in
pH, even if the composition is uniform over the same scales (Battaglia et al., 2017). Relative
humidity (RH) regulates aerosol liquid water content (ALWC), which affects the partitioning of
soluble gases and aqueous phase solute concentrations. ALWC may be the most important
factor responsible for large pH differences observed between the southeast US (pH ~0.5-1.0) and
the heavily polluted North China Plain (pH ~4-5), the two regions where pH has been most
extensively studied to-date (Zheng et al., 2020). These effects can be complex, or even partially
offset. For example, an increase in temperature reduces pH owing to the shift in $NH_3$
partitioning towards the gas phase, but $NH_3$ emissions increase with temperature as well,
producing an increase in pH and partially offsetting the pH changes due to temperature (Tao,
2020). More research is needed to better understand how all these factors together affect pH, and
how this changes geographically and temporally.



In this study aerosol pH was characterized during the summertime (June 4 to July 5,
2018) at a land-water transition site near a large urban area (Baltimore, MD) as part of the
OWLETS-2 (second Ozone Water-Land Environmental Transition Study) field campaign.  This
site is unique because meteorological phenomena, such as the bay breeze, affect pollution
dispersion and recirculation (Loughner et al., 2014).  Baltimore is impacted by different regional
emission sources, as it is located in the populous and heavily trafficked I-95 corridor, and
downwind of the Ohio River Valley (He et al., 2013), and relatively close to regional agricultural
operations that emit large amounts of $NH_3$ (Pinder et al., 2006).  In the companion investigation,
the local sources of $NH_3$ during the OWLETS-2 study were examined, including an analysis of
transient events with unexpectedly high $NH_3$ concentrations (Balasus et al., 2021).  In this study,
the effects of the observed $NH_3$ concentrations (including transient concentrations of
unexpectedly high values), in combination with the unique meteorological phenomena associated
with the land-water transition, on ALW and aerosol pH were investigated.

**2 Methods**
The OWLETS-2 study was conducted to characterize effects of meteorological
phenomena associated with the land-water transition on summertime air quality in Baltimore.
Hart-Miller Island (HMI, coordinates 39.2421°, -76.3627°), a site located on the Chesapeake Bay
~10 km east of downtown Baltimore, hosted many of the ground-based measurements during the
study (Fig. S1).  Semi-continuous measurements of aerosol inorganic chemical composition and
gas-phase $NH_3$ were conducted at HMI.  The measurement details are provided in the companion
paper (Balasus et al., 2021).  Briefly, the water-soluble ionic components of $PM_{2.5}$ were
measured with a Particle-into-Liquid Sampler coupled to a dual Ion Chromatograph (PILS-IC,





99 Metrohm) operated according to Valerino et al. (2017). $NH_3$ was measured with an AiRRmonia

100 Analyzer (RR Mechatronics) (Norman et al., 2009). Meteorological parameters were measured

101 with a Vaisala MAWS201 Met Station at 1-minute resolution.

102  The 5-minute $NH_3$ measurements and 1-minute meteorological measurements were

103 averaged to the 20-mintue sampling time of the PILS-IC. Aerosol pH for each 20-min sample

104 was calculated according to Zheng et al. (2020). This method uses the relevant temperature-

105 dependent equilibrium constants and the measured concentrations of $NH_3$ and $[NH_4^+]$ to

106 calculate pH. The method of Zheng et al. (2020) is based upon a similar approach in prior

107 studies (Hennigan et al., 2015; Keene et al., 2004), with minor differences possibly due to

108 equilibrium constant values and/or the concentration basis (molality vs. molarity). Indeed, very

109 close agreement was observed (slope = 0.967, $R^2$ = 0.977, n = 872, Fig. S2) in calculated pH

110 between the methods of Zheng et al. (2020) and Hennigan et al. (2015). The aqueous phase

111 $NH_4^+$ concentration is derived from the mass concentration of $NH_4^+$ from the PILS-IC and the

112 ALWC. The ISORROPIA-II thermodynamic equilibrium model was used to calculate ALW

113 using the PILS-IC, $NH_3$, and meteorological data as inputs (Fountoukis and Nenes, 2007). The

114 model was run in forward mode ($NH_3$ and aerosol $NH_4^+$ were input at total $NH_3$) using the

115 metastable assumption according to the recommendation of Guo et al. (2015).

116  Although ISORROPIA can provide pH, the methods of Zheng et al. (2020) and Tao et al.

117 (2020) were used for pH calculations:

$$pH = pK_{a,NH_3}^* + log_{10}\frac{[NH_3(aq)] + [NH_3(g)]}{[NH_4^+(aq)]} \tag{1}$$

$$[NH_3(g)] = \frac{p_{NH_3}\rho_w}{RT\,AWC} \tag{2}$$

$$K_{a,NH_3}^* = \frac{[H^+(aq)]([NH_3(aq)] + [NH_3(g)])}{[NH_4^+(aq)]} = K_{a,NH_3}\left(1 + \frac{\rho_w}{H_{NH_3}RT\,AWC}\right) \tag{3}$$



where in Equation 1, $[NH_3(aq)]$ is the molality of $NH_3$ in solution (mol kg$^{-1}$, calculated by
multiplying $H_{NH_3}$ and $p_{NH_3}$), $[NH_3(g)]$ is the equivalent molality of gaseous $NH_3$ in solution
(mol kg$^{-1}$, given by Equation 2), and $[NH_4^+(aq)]$ is the molality of $NH_4^+$ in solution (mol kg$^{-1}$).
Equation 2 calculates $[NH_3(g)]$, where $p_{NH_3}$ is the partial pressure of $NH_3$ (atm), $\rho_w$ is the
density of water (µg m$^{-3}$), R is the gas constant (atm L mol$^{-1}$ K$^{-1}$), T is temperature (K), and
AWC is aerosol water content (µg m$^{-3}$ air).  Equation 3 calculates the last unknown term in
Equation 1, which is $K_{a,NH_3}^*$, the effective dissociation constant (where $H_{NH_3}$ is the Henry's law
constant of $NH_3$ in mol kg$^{-1}$ atm$^{-1}$).

Direct measurements of aerosol pH are not available to test model predictions so the

partitioning of semi-volatile species that depend on pH, most commonly $NH_3/NH_4^+$ and
$HNO_3/NO_3^-$, is a key metric used to evaluate model performance (Pye et al., 2020).
ISORROPIA is used extensively for predictions of pH; however, in this study the measured and
ISORROPIA-predicted values of $NH_3$ partitioning ($\varepsilon_{NH3} = NH_3/(NH_3 + NH_4^+)$) did not agree well
(Fig. S3).  There were systematic differences in pH between the two methods (mean pH
difference = 0.6 pH units), and they were not correlated ($R^2$ = 0.097, not shown).  The source of
the discrepancy in $NH_3$ partitioning (ISORROPIA modeled versus measurement-calculated) is
unknown, though it is not likely the result of an incorrect assumption of equilibrium (see the SI
and the companion paper Balasus et al. (2021) for more details).

**3 Results and Discussion**
**3.1 Meteorological effect on pH**

As in many cities, the urban heat island effect in Baltimore evolves throughout the day,

with urban-rural temperature and RH gradients peaking at night (Battaglia et al., 2017).



Meteorological conditions at HMI demonstrate unique features associated with the land-water
transition.  At night, the average temperature observed at HMI was close to conditions observed
in downtown Baltimore but transitioned to match the conditions at a nearby rural site during the
day (Fig. 1a). Overall, the range in average hourly temperatures at HMI was lower (4.3 °C; 22.5
– 26.8 °C) than the averages observed at either the downtown site (range 6.2 °C; 22.8 – 29.0 °C)
or the rural site (range 8.8 °C; 17.5 – 26.3 °C).  Likewise, the average hourly RH profile at HMI
had a significantly smaller range (13.7%; 61.7 – 75.4%) than the RH profiles at the downtown
(range 22.1%; 53.2 – 75.3%) or rural sites (range 30.3%; 55.9 – 86.2%) (Fig. 1b).  The
differences shown in Fig. 1 were due to the proximity of HMI to the Chesapeake Bay.  This has
strong implications for aerosol pH, which will be discussed in detail below.

Due to the meteorological conditions discussed above, the diurnal profile of ALWC at

HMI was unique (Fig. 2a).  Typical profiles of ALWC in the eastern US closely follow RH, with
minima in the afternoon and maxima at night or in the pre-dawn morning hours (Guo et al.,
2015; Battaglia et al., 2017).  During OWLETS-2, ALWC did not show a distinct diurnal profile
that was correlated with RH.  Instead, the highest median ALWC at HMI occurred between
12:00 – 14:00, (local time, LT; during the study this is UTC-4) (Fig. 2a).  This daily peak in
ALWC coincided with a pronounced enhancement in aerosol $NO_3^-$, which is discussed in the
companion paper (Balasus et al., 2021).  The partitioning of $NH_3$, $\varepsilon_{NH3}$, also showed a diurnal
profile that was unexpected (Fig. 2b).  Due to the strong temperature dependence of vapor
pressure and equilibrium constants, $NH_3$ partitioning typically shifts towards the gas-phase
during the daytime ($\varepsilon_{NH3}$ increases) and shifts towards the aerosol phase at night ($\varepsilon_{NH3}$ decreases)
(Guo et al., 2017).  The increase in gas-phase $NH_3$ emissions with increasing temperature can
also contribute to an elevated $\varepsilon_{NH3}$ during the daytime.  The diurnal profile of $\varepsilon_{NH3}$ observed at



HMI did not follow temperature, as the median $\varepsilon_{NH_3}$ peaked during the 06:00 – 08:00 LT, and
decreased slightly into the afternoon (Fig. 2b).  This shows a shift of NH$_3$ partitioning towards
the condensed phase as daily temperatures peaked.  This was allowed by the ALWC remaining
steady throughout the afternoon.  Overall, the median $\varepsilon_{NH_3}$ value for the entire OWLETS-2 study
was 0.915, showing NH$_3$ partitioning was shifted towards the gas-phase.

The diurnal profile of aerosol pH computed using the method of Zheng et al. (2020)

followed a qualitatively similar pattern to prior studies (Battaglia et al. 2017), with maxima in
the early morning and minima in the afternoon; however, there was a much smaller amplitude in
the median hourly pH values (Fig. 3).  The highest median pH value (1.97) was observed
between 07:00 – 08:00 LT, while the lowest median pH (1.50) was observed between 16:00 –
17:00 LT.  For the entire study, there was only a ~1 pH unit difference between the 10[th] and 90[th]
percentile values (1.39 and 2.36, respectively).  The relatively muted diurnal profile of aerosol
pH was due to the unique meteorology that resembled a nearby urban site at night and
transitioned to match the nearby rural site during the day.  The companion paper shows the
diurnal profiles of aerosol inorganic composition (Balasus et al., 2021).  Figures 2 and 3 are
consistent with recent studies that demonstrate the high sensitivity of pH to meteorological
factors (Battaglia et al., 2017; Tao and Murphy, 2019; Zheng et al., 2020).  It is interesting to
note that the ISORROPIA predictions of aerosol pH yield a distinctly different diurnal profile
than the pH predicted by NH$_3$ partitioning (Fig. S4).  The difference in pH between the two
methods peaked in the afternoon between 12:00 – 14:00 LT, when particles were most acidic,
and was a minimum in the early morning when pH was highest (Fig. S4).  The reason for these
discrepancies are explored in the discussion below.



Results suggest that pH was most sensitive to temperature changes at HMI during
OWLETS-2 (Fig. 4a).  Across the full temperature range, the observed pH sensitivity to
temperature was -0.047 K$^{-1}$.  A recent study contrasting pH in the southeast US and the North
China Plain found that temperature affects pH linearly at a rate of approximately -0.055 K$^{-1}$
(Zheng et al., 2020).  A separate study from a Canadian observational network found that the
pH-temperature dependence is not linear, but changes with temperature (Tao and Murphy, 2019;
Tao, 2020).  Over the range of conditions observed during OWLETS-2, Tao (2020) computed a
pH sensitivity to temperature of approximately -0.045 K$^{-1}$ to -0.055 K$^{-1}$.  In a previous study, the
pH sensitivity to temperature in Baltimore was -0.048 K$^{-1}$, though this was calculated for
conditions of constant atmospheric composition (Battaglia et al., 2017).  While present results
share consistencies with these studies, the results in Fig. 4a suggest important differences, as
well.  An increase in pH was observed with increasing temperature for conditions below 293 K
(n=156).  In the companion paper, NH$_3$ concentrations dramatically increased with temperature <
293 K but exhibited a much weaker dependence on temperature for conditions > 293 K (Balasus
et al., 2021).  Under the warmer conditions, the pH relationship with temperature was linear
during OWLETS-2, with a sensitivity of -0.085 K$^{-1}$ (Fig. 4a).  The results in Fig. 4a are
consistent with Tao (2020), as they demonstrate the offsetting responses of pH to temperature
through effects on NH$_3$ emissions and partitioning.
These results contribute to the growing body of work demonstrating the importance of
temperature to aerosol pH.  Collectively, these studies suggest that the sensitivity of pH to
temperature is constrained between -0.045 K$^{-1}$ and -0.085 K$^{-1}$, although the present results
illustrate circumstances where a positive relationship between temperature and pH can exist, as
well.  It is notable that similar $\Delta$pH/$\Delta$T values are observed across a range of locations and for



variable data sets that include monthly averages of long-term observations (Tao and Murphy,
2019) and 20-min measurements made over a period of weeks (presented here).
Aerosol pH was not strongly affected by ALWC over the range of conditions observed
during OWLETS-2, except at ALWC below 1 μg m⁻³ (Fig. 4b). At the lowest ALWC levels, the
increase in pH occurs because of the diluting effect of water, however, at ALWC above 1 μg m⁻³,
other factors appear to be more important (e.g., temperature). ALWC was recently identified as
the most significant contributor to regional pH differences (Zheng et al., 2020). In that study, the
pH-ALWC relationship was highly non-linear, with the greatest sensitivity calculated at ALWC
at levels < 25 μg m⁻³, conditions corresponding to all the OWLETS-2 observations. Tao (2020)
found that pH is extremely sensitive to RH, presumed to be a surrogate for ALWC, when RH
was < 20% or RH > 80%; however, pH was quite insensitive to RH variations in the region of
20% < RH < 80%. Approximately 25% (208 out of 875) of RH values during OWLETS-2 were
above 80%, yet no increase in pH was observed at the highest ALWC, suggesting that other
factors were offsetting the diluting effect of water as ALWC increased. It is interesting to note
that ISORROPIA predicts a stronger effect of ALWC on pH, with the diluting effect apparent as
pH increases with increasing ALWC (Fig. 4b).
The result shown in Fig. 4b is somewhat surprising because $NH_3$ partitioning was quite
sensitive to ALWC (Fig. 5); the relatively invariant aerosol pH is unexpected given the increase
in $NH_3$ uptake in the presence of ALW. $NH_3$ partitioning shifted towards the condensed phase
($\varepsilon_{NH3}$ decreased) at increasing ALWC, consistent with the results of Nenes et al. (2020). $\varepsilon_{NH3}$
was more sensitive to ALWC than it was to either temp or RH (Fig. S6). This result shows the
importance that ALWC can have on $PM_{2.5}$ mass concentrations, as water serves as an important
medium enhancing the condensation of organic and inorganic water-soluble species (Carlton et





al., 2018; El-Sayed et al., 2016). Fig. 5 demonstrates the importance of ALWC in changing the
dry deposition of reactive nitrogen species. Nenes et al. (2021) predicts that $NH_3$ and $HNO_3$ dry
deposition rates will both be high under the conditions observed during OWLETS-2 (i.e., ALWC
< 10 µg m$^{-3}$ and pH ~ 1.5). This is also consistent with a modeling study showing increased dry
deposition of reactive nitrogen in coastal regions, including the OWLETS-2 study domain
(Loughner et al., 2016).

**3.2 Composition Effects on Aerosol pH**

In contrast to meteorological factors, aerosol composition did not have a major effect on

pH variability during OWLETS-2. Consistent with prior studies, neither the sulfate
concentration nor the $NH_4^+$:$SO_4^{2-}$ molar ratio contributed significantly to pH variability (Weber
et al., 2016; Hennigan et al., 2015). Fig. 4c shows that pH was also relatively insensitive to the
Tot-$NH_3$ concentration, in agreement with the results from other locations (Zheng et al., 2020;
Tao, 2020; Weber et al., 2016). While the predictions of ISORROPIA generally did not capture
the trends between pH and the meteorological factors, it is interesting to note that ISORROPIA
predicts that pH is relatively insensitive to Tot-$NH_3$, as well, except at the highest concentrations
(Fig. S5).

Composition and concentration differences between HMI and the urban and rural sites

are analyzed in more detail in the companion paper (Balasus et al., 2021). In the companion
paper, episodic $NH_3$ events that derived from dairy, poultry, and industrial sources were
characterized (Balasus et al., 2021). For the events with complete aerosol composition and
meteorology data (8 out of 11 $NH_3$ events total), the average and medial aerosol pH values (2.00
and 1.96, respectively) were only moderately higher than the study average and median pH





values (1.85 and 1.83, respectively); this difference is not statistically significant at the 95%
confidence interval.  This includes an aerosol pH of 1.92 at the peak $NH_3$ concentration observed
during the entire study (19.3 µg m$^{-3}$), an event influenced by industrial emissions near downtown
Baltimore (Balasus et al., 2021).  Together, the results suggest that pH may be more affected by
the proximity of the HMI site to the Chesapeake Bay than it was to regional agricultural $NH_3$
emissions, or to episodic $NH_3$ events from local industrial sources.

**3.3 Case Study: Effect of NVCs on pH**

NVCs affect thermodynamic predictions of $NH_3$ partitioning and thus, pH (Guo et al.,

2018; Vasilakos et al., 2018).  Seawater is alkaline and primary marine emissions contain high
concentrations of NVCs (O'Dowd and De Leeuw, 2007).  Marine aerosols rapidly acidify,
typically in seconds or minutes, though the timescale depends upon particle size (Angle et al.,
2021; Pszenny et al., 2004).  Studies have examined the acidity of sea spray aerosols and their
evolution but none, to the authors' knowledge, have done so in a polluted urban environment.
The OWLETS-2 study offered a unique opportunity to analyze the pH of primary marine
particles emitted within several km of a large urban area.  The Chesapeake Bay is brackish, with
increasing salinity moving down the bay towards the Atlantic Ocean (Pritchard, 1952).  Near the
OWLETS-2 measurement site at HMI, salinity is variable but average conditions are ~5 g kg$^{-1}$
(www.chesapeakebay.net, last accessed 20-November 2020).  This is about a factor of seven
lower than typical seawater, but shows the potential for primary emissions to contribute salts that
could impact aerosol pH at HMI.

Elevated concentrations of $Na^+$ and $Cl^-$ in $PM_{2.5}$ were infrequently observed during

OWLETS-2, with one notable 36-hour period showing evidence of primary marine impact.  $Na^+$



and $Cl^-$ were well-correlated ($R^2 = 0.78$) from 11-June to 13-June, a period that coincided with
the highest concentrations of both species (Fig. 6 and Fig. 7). It is noteworthy that wind speeds
were not elevated during this time (average winds = 3.1 m s$^{-1}$ compared to campaign-average
wind speeds of 2.9 m s$^{-1}$); longer-term measurements would be needed to characterize factors
driving the primary bay emissions. During this event, the pH calculation using $NH_3$ partitioning
suggests that the primary marine emissions had a minimal effect on aerosol pH. The average pH
during this period was 1.98, which was only slightly higher than the average for the entire study
(1.85). Further, as the total $Na^+ + Cl^-$ concentration increased by more than an order of
magnitude, from 0.05 μg m$^{-3}$ around 18:00 LT on 11 June to 0.9 μg m$^{-3}$ at 09:00 LT on 12 June,
the pH only increased by 0.1 pH unit during the same period (Fig. 6). Gas-phase $NH_3$ data were
not available for the entire NaCl event, so pH calculations were limited to the first ~20 hours.

The pH predictions from ISORROPIA during this period display a more significant effect

on aerosol pH. The average pH predicted by ISORROPIA is 3.08, which is 0.7 pH units higher
than the study average from ISORROPIA. Further, while the pH calculated using $NH_3$
partitioning is insensitive to $Na^+$ and $Cl^-$, ISORROPIA predicts a rise of 1.4 pH units as $Na^+$ and
$Cl^-$ increase (Figure 6). The $NH_3$ partitioning predicted by ISORROPIA deviates from the
observations during this event (r = -0.19, Fig. S7). The most likely explanation for this behavior
is different chemical compositions of the coarse- and fine-mode particles. Fresh marine
emissions acidify quickly, and evidence was found for chemical processing of NaCl. $HNO_3$
displacement of HCl is a well-known phenomenon in sea salt particles (Brimblecombe and
Clegg, 1988). Chloride:sodium ratios slightly above unity were observed when aerosol nitrate
concentrations were low, and below unity when nitrate concentrations were elevated (Fig. 7).
Nitrate formation, including $HNO_3$ uptake to sea salt, is highly sensitive to pH (Kakavas et al.,
2021; Vasilakos et al., 2018).  Nitrate (along with $Na^+$ and $Cl^-$) are direct inputs to ISORROPIA,
suggesting that the pH trend in Fig. 6 is due to the increased influence of coarse-mode particles.
The PILS inlet was equipped with a 2.5 μm cut-point cyclone (URG-2000-30-EH, URG Corp.),
which allows some penetration of particles with $d_p$ < 4.5 μm
(http://www.urgcorp.com/products/inlets/teflon-coated-aluminum-cyclones/urg-2000-30eh, last
accessed 29 January 2021).  Likewise, primary sea salt emissions often exhibit a size distribution
tail that extends below 2.5 μm (Feng et al., 2017).

The above analysis identifies limitations computing aerosol pH with both approaches and

highlights opportunities to use complementary information from each to inform factors driving
aerosol pH.  ISORROPIA's assumption of an internally mixed aerosol distribution cannot be
applied to predict pH in this system (Fountoukis and Nenes, 2007).  However, it does provide
insight into the likely presence of coarse mode particles with pH significantly higher than the
fine mode.  Conversely, $Na^+$, $Cl^-$ and $NO_3^-$ are not direct inputs into the pH calculation using
$NH_3$ partitioning, though they are used to compute aerosol liquid water, which is an input in the
pH calculation (Zheng et al., 2020).  These NVCs have been shown to have sometimes
significant effects on the ratios of ammonium-sulfate, which could lead to inconsistencies in the
calculation of aerosol pH when considered in the ALWC calculation only (Guo et al. 2015).
$NH_4^+$ resides predominantly in the fine mode (Seinfeld and Pandis, 2016), so the partitioning
approach is unlikely to capture the acidity of the coarse mode or fine-mode particles in the tail of
the distribution of primary emissions that may be externally mixed with secondary particles, such
as dust or sea salt.  This may lead to underestimates of NVC effects on aerosol pH using the
partitioning approach.  The combined information from both methods suggests that aerosols
sampled at HMI during the NaCl event were characterized by a strong size-dependent pH





gradient, with fine-mode particles more acidic (pH ~2) than the coarse mode (pH up to 4.5).
Estimates of pH derived from size-segregated aerosol composition measurements have observed
the same phenomenon (Angle et al., 2021; Fang et al., 2017; Kakavas et al., 2021; Keene et al.,

2002).


**4 Conclusions**

There is growing recognition of the importance of aerosol pH affecting atmospheric

processes relevant to public health and ecosystems (Fang et al., 2017; Nenes et al., 2021).
Observations of spatial and temporal variations in pH are needed so that the factors that control
pH and contribute to variability in different environments can be fully understood. This study is
unique as it represents the first characterization of aerosol pH at a land-water transition site near
a large urban area. Baltimore, MD, is impacted by regional agricultural emissions and by
industrial point-source emissions of $NH_3$. The companion paper examines the sources of
episodic $NH_3$ events and the associated effects on aerosol composition (Balasus et al., 2021).
Although average and peak $NH_3$ concentrations during this study were significantly higher than a
nearby inland site, the effects on aerosol pH appear relatively insignificant, as pH during the
peak events was only ~0.1 pH unit higher than non-event periods. This finding is consistent with
studies at other locations that show aerosol pH is often insensitive to Tot-$NH_3$ and to the aerosol
$NH_4:SO_4$ ratio (Weber et al., 2016; Zheng et al., 2020; Tao and Murphy, 2019).

The unique characteristics of the OWLETS-2 study and measurement locations also

offered insight into the composition and meteorological influences on aerosol pH. In the
companion paper, the composition effects were shown to be muted in comparison to the
meteorological effects (Balasus et al., 2021). It was shown that the unique diurnal profiles,




particularly in ALW (which did not correlate with RH) and $\varepsilon_{NH3}$ (which did not correlate with T)
resulted in meteorological factors, notably temperature, having the most important influence on
aerosol pH.  Across the full temperature range of the study, the observed pH sensitivity to
temperature was -0.047 K$^{-1}$, with increases in sensitivity up to -0.085 K$^{-1}$ when the temperature
was > 293 K.  The sensitivity of aerosol pH shown here is in good agreement with previous
studies in the Baltimore region and beyond, *e.g.* Toronto (Tao and Murphy, 2019; Battaglia et al.
2017).  Conversely, aerosol pH was not strongly affected by ALWC during the OWLETS-2
study, except when ALWC was below 1 µg m$^{-3}$, in contrast to the results of Zheng et al. (2020).
These results of Zheng et al. (2020) in identifying ALW as the most important factor driving
aerosol pH variability were derived from the results of studies in multiple locations, including
bulk aqueous solution.  However, the analysis suggests that the factors that drive aerosol pH
variability may exhibit important site-to-site differences that must be considered before
generalizations are applied.

A case study of the NVC effect on aerosol pH had significantly different outcomes

depending on the method for calculating pH.  ISORROPIA predicted a pH increase of ~ 1.4 pH
units during an event with primary aerosol emissions from the Chesapeake Bay, while the pH
calculation using NH$_3$ partitioning predicted a much less significant effect (~0.1 pH unit).  This
difference is attributed to the likely presence of externally mixed particles during the events,
which may include primary marine emissions elevated in NVCs.  Bougiatioti et al. (2016)
evaluated aerosol pH at a remote site in the Mediterranean, where samples with a marine origin
demonstrated vastly different pH between fine (avg. pH = 0.4) and coarse mode (average pH =
7.3) particles.  Similarly, Keene et al. (2002) demonstrated the effect of marine aerosol size
distribution on aerosol pH, with fine mode particles predicted to reside in the range of 1-2, with



super-μm particles to reside in the range of 3-4, consistent with the current results. Hence, a
limitation of this study is the lack of size-resolved aerosol composition measurements. This
study underscores the need to evaluate assumptions of internally mixed aerosols when applying
pH calculations, which may be a critical factor in overestimating the effects of NVCs on pH.
Models with size-resolved aerosol composition may be required to capture this effect across
scales in future studies (Kakavas et al., 2021).

**Data Availability**
Data are available at https://www-air.larc.nasa.gov/cgi-bin/ArcView/owlets.2018.

**Supplement**
The supplement related to this article is available online at:

**Acknowledgements**
A.G.C. and C.J.H. acknowledge funding from the National Science Foundation, AGS-1719252
and AGS-1719245. R.D. and V.C. acknowledge support by the National Oceanic and
Atmospheric Administration – Cooperative Science Center for Earth System Sciences and
Remote Sensing Technologies under the Cooperative Agreement Grant #: NA16SEC4810008.
N.B. and K.B. received support through the NOAA Office of Education, Educational Partnership
Program with Minority Serving Institutions (EPP/MSI).

**Author Contributions**





CH, AC, and RD conceived the analysis and study participation. MB, NB, and KB collected and
analyzed the PILS-IC and $NH_3$ data. NB, MB, and CH conducted the thermodynamic modeling
analyses. VC and AC provided analytical input and interpretation. NB, CH, KB, and MB wrote
the manuscript. All authors provided feedback and revisions to the manuscript.

**Competing Interests**
The authors declare they have no conflict of interest.





**Figures**

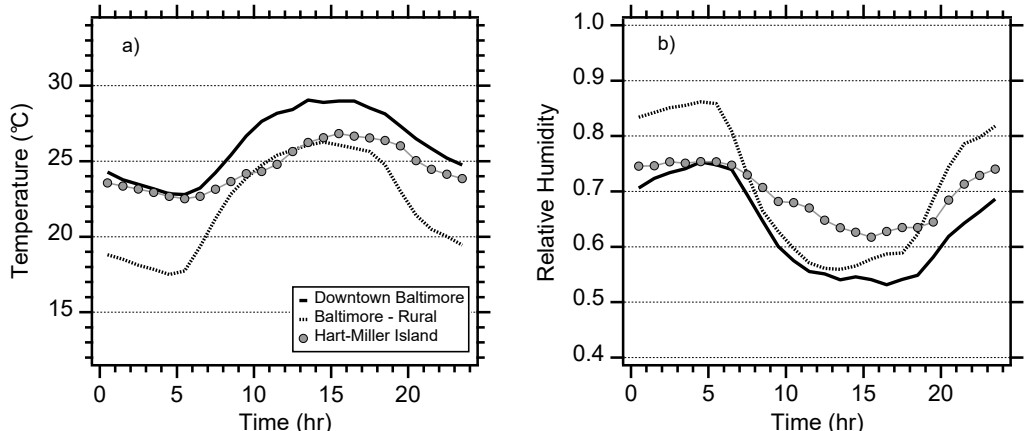


**Figure 1:** Diurnal profiles of (a) temperature and (b) relative humidity at three sites during the
OWLETS-2 study.  Hart-Miller Island (HMI) is a land-water transition site and was the location
of aerosol composition and gas-phase measurements during the campaign.  Temperatures at HMI
resembled the downtown location during the night but showed characteristics of the rural site
during the day.  RH at HMI was between the urban and rural sites at night but was elevated
during the daytime due to the Chesapeake Bay.

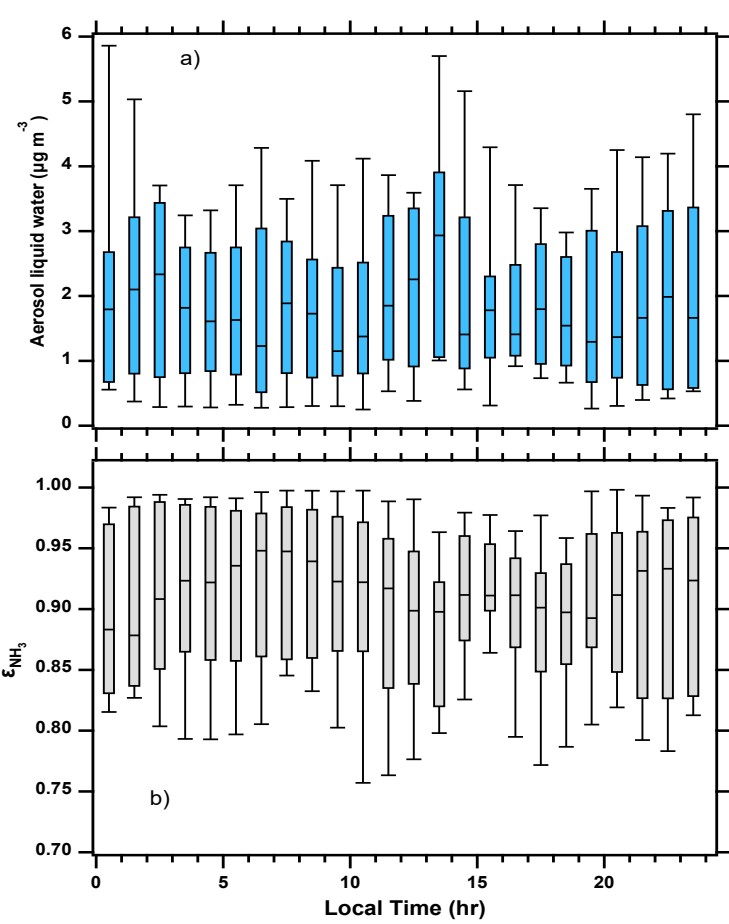


**Figure 2:** Box plots of (a) aerosol liquid water and (b) $\varepsilon_{NH3}$ ($\varepsilon_{NH3}$ = NH$_3$(g)/(NH$_3$(g) + NH$_4^+$(aq)))

during the OWLETS-2 study. The statistics shown are the median, quartiles, 10$^{th}$ and 90$^{th}$

percentiles.



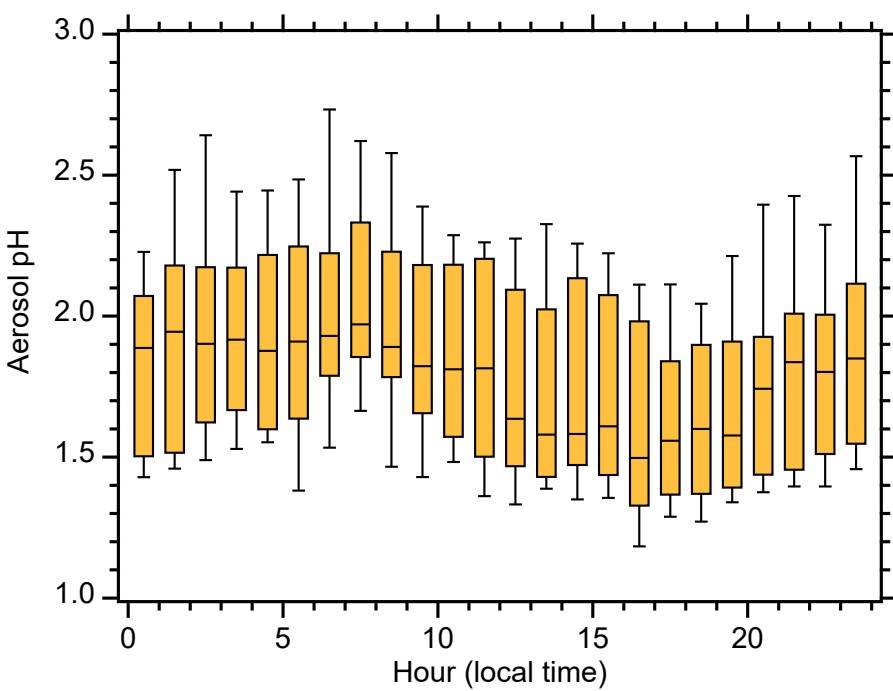


**Figure 3:** Box plot of the aerosol pH diurnal profile calculated using the method of Zheng et al.

(2020) at Hart-Miller Island during OWLETS-2. The statistics shown are the median, quartiles,

10[th] and 90[th] percentiles.

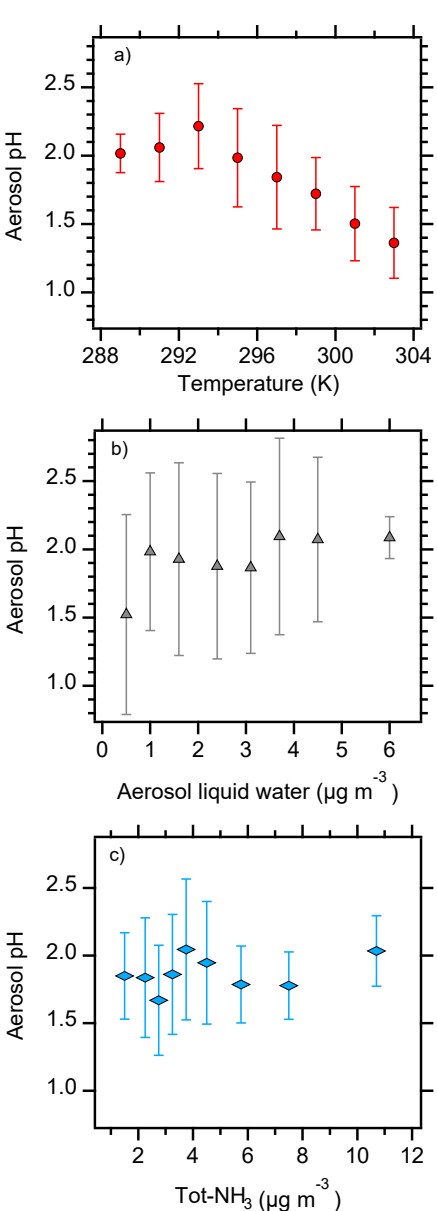


**Figure 4:** Relationship between aerosol pH and (a) temperature, (b) aerosol liquid water, and (c)

total $NH_3$ (Tot-$NH_3$ = $NH_3$ + $NH_4^+$).  Symbols represent mean values while error bars represent

standard deviations.





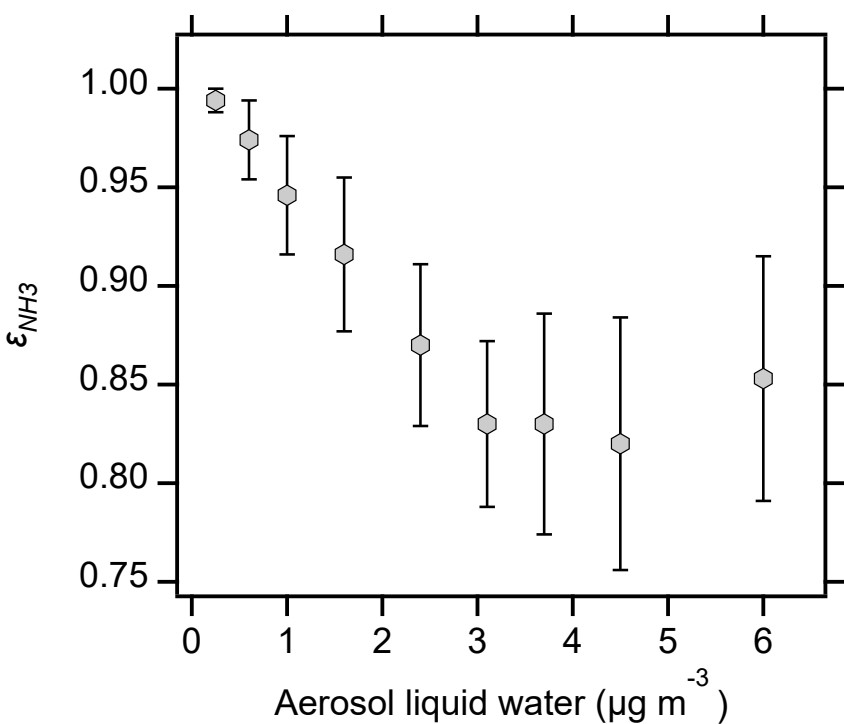


**Figure 5:** Relationship between NH₃ partitioning ($\varepsilon_{NH3}$ = NH₃/(NH₃ + NH₄⁺)) and aerosol liquid

water. Symbols represent mean values while error bars represent standard deviations.




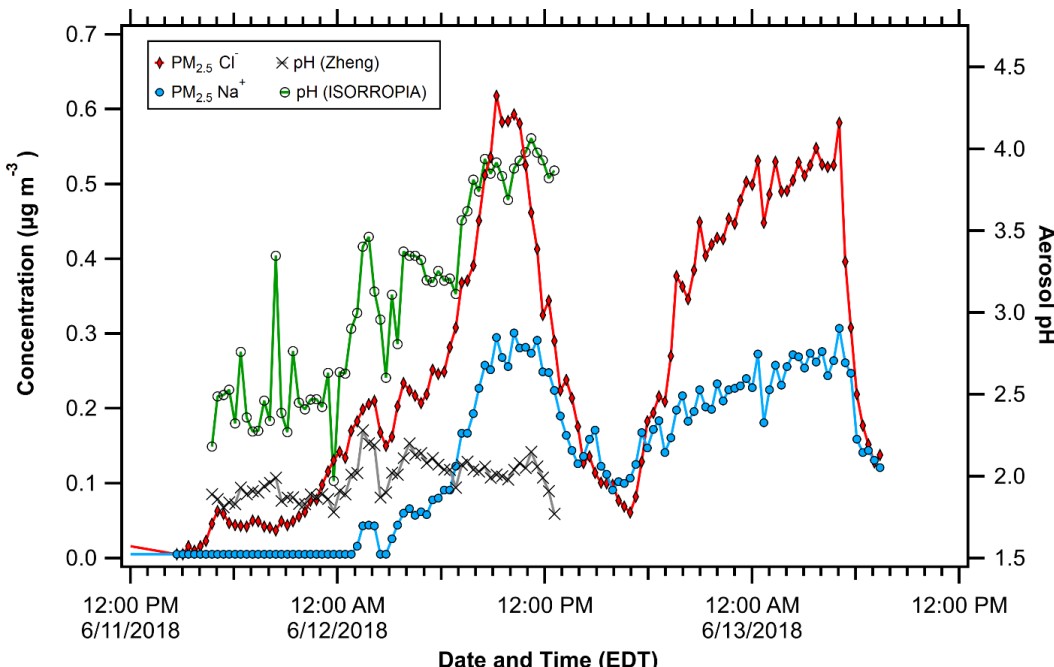


**Figure 6:** Concentrations of PM$_{2.5}$ Na$^+$ and Cl$^-$ during a period with primary Chesapeake Bay

emissions. Aerosol pH calculated by NH$_3$ partitioning (Zheng et al. method) and ISORROPIA
show different behaviors, suggesting the aerosol distribution was externally mixed during this
time.



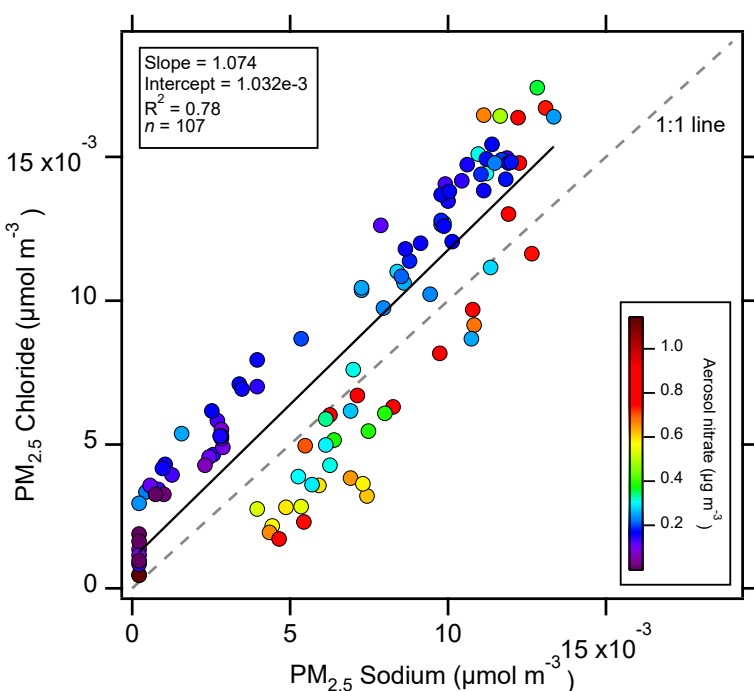


**Figure 7:** Correlation between PM$_{2.5}$ Na$^+$ and Cl$^-$ during the event shown in Figure 6. Lower

chloride:sodium ratios were observed at higher NO$_3^-$ concentrations, suggesting HNO$_3$ uptake

displaced HCl.





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
