# Peer review of "Urban aerosol chemistry at a land-water transition site during summer – Part 2: Aerosol pH and liquid water content"

_Atmospheric Chemistry and Physics, 2021_

## Author Response (AR1)

*Response to Reviewer Comments for "Urban aerosol chemistry at a land-water transition site during summer – Part 2: Aerosol pH and liquid water content" by Michael A. Battaglia Jr. et al., Atmos. Chem. Phys. Discuss., https://doi.org/10.5194/acp-2021-368-RC1, 2021"*

We thank the reviewers for their detailed and helpful comments.  We have addressed each comment with the Referee comments in bold and our reply in plain text immediately below.  We provide at the end the edited manuscript with all changes highlighted.

**REVIEWER #1**

**Major Comments**

**1.) It looks necessary to quantify the impact of measurement uncertainties to these two pH calculation methods. The authors should also provide the QA/QC information, not only limited to concentrations, but also the impact on possible ranges of pH calculated.**

We have attempted to address this issue by referring readers to the literature.  Additional discussion of the impact on calculated aerosol pH based on uncertainty in the measured input concentrations have been added.  The following discussion was added at the conclusion of the Methods section of the paper to elaborate on these uncertainties:

"Commonly, a reasonable estimate of measurement errors associated with the species of interest are on the order of 15% + 1 nmol m$^{-3}$ for online sampling methods (Murphy et al. 2017). Utilizing this estimate in the extreme acidic case (ambient observed $SO_4^{2-}$ and $NO_3^-$ adjusted to + 15% + 1 nmol m$^{-3}$; $NH_4^+$ adjusted to – 15% – 1 nmol m$^{-3}$) or non-acidic case ($SO_4^{2-}$ and $NO_3^-$ adjusted down by – 15% – 1 nmol m$^{-3}$ and $NH_4^+$ adjusted up by + 15% + 1 nmol m$^{-3}$), the uncertainty of aerosol pH was predicted to vary by 0.1 – 1 pH units (Murphy et al. 2017). Similarly, Pye et al. (2020) reported that in cases for RH above 60% deviations of the ISORROPIA-predicted pH and the IUPAC-defined pH are less than one pH unit. The present calculations have similar uncertainties in the pH calculations."

**2.) Line 107-108: Is it because how pH was defined (e.g. pH$_c$ or pH$_m$ in Jia et al. (2018))?  It's better to clarify how the pH was defined in the first place.**

This statement has been clarified to reflect that the methods of Hennigan et al. 2015 and Zheng et al. 2020 are identical with the exception of the values of the equilibrium constants used. Based on our investigation, the non-ideality corrections introduced in Zheng et al. 2021 appear to be working towards ISORROPIA, but instead of calculating activity coefficients they interpolate with thermodynamic outputs from ISORROPIA (or E-AIM if desired) to try to mimic the thermodynamic models based on what they say impacts non-ideality the most (they base it on T, RH, and $NO_3^-$). The text has been updated to read:

"The method of Zheng et al. (2020) is based upon a similar approach in prior studies (Hennigan et al., 2015; Keene et al., 2004); the two methods are identical, with the minor differences a result of equilibrium constant values used based on ranges provided in the literature."

3.) **Line 153: Looks like the authors intend to make comparisons of pH in different locations and discuss the influence from RH and T, but I did not see such discussions in the following discussion. I think the authors should clarify what "strong implications" they are referring to.**

The differences between locations were intended to describe the changes in diurnal T, RH, and ALWC between the Baltimore (urban) and HMI sites. The unique character of these profiles at the HMI site (neither strongly urban nor rural) was investigated as a driver of the model-predicted aerosol pH during the OWLETS 2 study. The discussion points have been linked together, and the original statement on Line 153 has been changed to read:

"The differences shown in Fig. 1 were due to the proximity of HMI to the Chesapeake Bay and have several implications for aerosol pH, which will be discussed in detail below."

4.) **Line 162-171: NH3 phase partitioning also strongly depends on the availability of accompanied acids. Without balance ions, NH3 alone cannot form particles. On the other hand, for pure ammonium sulfate, the evaporation tendency of NH3 is also quite limited. Low concentrations of particulate acids could also lead to ammonia-rich atmosphere.**

We have attempted to address this comment together with Reviewer 2 Major Comment #2. Please see the response to Reviewer 2 Major Comment #2 for details.

5.) ***Figure 4 &5: These two figures provide valuable local information about pH response and of high importance for similar studies comparison. As a result, the authors should provide a more complete description about how the data points are chosen, averaged or processed. The authors have 20-minutes resolution data, while only 8 points are plotted. How were the bins chosen and why not just use all the measured data?***

Figures 4 and 5 have been expanded to display both the binned data (from the original manuscript) together with the entire collection of raw unbinned data used to generation the original figures to show any relevant trends. We have clarified the binning process by describing the maximum number of points permitted in each bin by addition of the following text to the figure captions:

"**Figure 4:** Relationship between aerosol pH and (a) temperature, (b) aerosol liquid water, and (c) total $NH_3$ (Tot-$NH_3$ = $NH_3$ + $NH_4^+$). Symbols represent mean values while error bars represent standard deviations; bins were limited to a maximum of 160 points per bin."

"**Figure 5:** Relationship between $NH_3$ partitioning ($\varepsilon_{NH3}$ = $NH_3$/($NH_3$ + $NH_4^+$)) and aerosol liquid water. Symbols represent mean values while error bars represent standard deviations,; bins were limited to a maximum of 160 points per bin."

6.) ***Line 225-231: Either high RH or high particle mass loading can be responsible for high ALWC, while it's hard to say the later case corresponds to the dilution effect. Could the authors distinguish which one dominates the ALWC? In the latter discussion, the authors mentioned the particles were likely externally mixed, so that the ALWC involved in the $NH_3$ phase partitioning processes should be, more or less, overestimated. Should that be considered here as well?***

Reviewer 2 expressed a related sentiment in Minor Comment #5 (Lines 228-240 of the original manuscript). While the additional discussion of that comment addresses, more specifically the surprising result of seemingly contradictory findings on relatively invariant pH with both increasing NH$_3$ and ALWC, we have also added clarifying statements to address this comment as well. The following comment has been added at the end of line 227 of the original manuscript:

"A combination of particle mass loading, aerosol composition, and ambient RH are responsible for the variations in ALWC."

Additionally, it should be noted that previous studies have shown that the ALWC predictions of aerosol thermodynamic equilibrium models are highly accurate. For example, Guo et al. (2015) observed excellent agreement between model-predicted ALWC and direct observations during the SOAS campaign. This is one of the key parameters evaluated when a new model is published (e.g., Fountoukis and Nenes, 2007). This is further supported by the results of Ansari and Pandis (1999), who found generally excellent agreement in ALWC predictions between four different thermodynamic equilibrium models, even when predictions of other semivolatile species did not agree.

**Minor Comments**

*1.) Introduction: it's better to mention some current existing methods of pH direct Measurement.*

We have added the following discussion regarding the existing method of aerosol pH direct measurement:

"However, direct measurements of aerosol pH remain challenging. Single particle studies of aerosol pH using Raman spectroscopy have been performed, but are limited by the presence of both HSO$_4^-$ and SO$_4^-$ limiting their application to more acidic particles (Boyer et al., 2020; Rindelaub et al., 2016). Colorimetric measurements of aerosol pH have also been employed, but such techniques have been limited to laboratory studies with relatively simple aerosol compositions (Craig et al., 2018; Jang et al., 2020)."

*2.) Line 49: I do not follow the logic here, how does the previous sentence explain the later?*

The sentences were intended to be linked based on the controlling regimes of $NH_3$ influence on aerosol pH, *i.e.*, the compositionally-controlled or meteorologically-controlled. The statements have been linked and clarified in the following manner and the text now reads:

"$NH_3$ partitioning is controlled by the concentration of strong acids and by the ambient temperature and relative humidity, hence, the dependence of pH on both composition and meteorology (Zheng et al., 2020). In some regimes, the meteorological factors are more important than the compositional factors, explaining why $NH_3$ can exist partially in the gas phase even when the aerosol pH is highly acidic (Nenes et al., 2020; Weber et al., 2016)."

**3.) *Figure 7: Should the relative concentrations of nitrate be more reflective to the Cl displacement extent than absolute concentrations? And on this Figure, about half the data points have Cl:Na >1, what's the possible explanation?***

We do not feel that the relative concentrations of nitrate are more reflective to the Cl displacement extent than absolute concentrations. The additional components of sampled PM mass may have effects on the relative concentrations, but are unlikely to play a part in the actual displacement reaction. Regarding the Cl:Na > 1 condition present, our expectation is that there is an additional source of Cl that is difficult to account for, such as combustion sources from the similar industrial point sources documented in the Part 1 manuscript, local boat traffic, or equipment and truck traffic associated with work activities on Hart-Miller Island. Because we do not have a way to identify the additional Cl source, such discussion would be speculative.

**REVIEWER #2 RESPONSES**

**Major Comments**

1.) *Given the poor performance of ISORROPIA for NH3 partitioning predictions (pH dependent) in this study, Eq. 1-Eq. 3 was used to calculate aerosol pH, and the pH calculated from these equations was significantly different from the ISORROPIA-predicted aerosol pH. However, these equations are for ideal conditions (without considering ion activity coefficients), and the non-ideality in aerosols can introduce deviations from the ideal conditions. Zheng et al. (https://doi.org/10.5194/acp-2021-55) has recently introduced a non-ideality correction factor for using these equations to calculate aerosol pH, and the aerosol pH calculated from the non-ideality corrected equations agreed well with the pH value determined by ISORROPIA.*

*Therefore, I suggest the authors to use the non-ideality corrected equations (either with non-ideality correction factor or with the related ion activity coefficients) to calculate aerosol pH and then compare with ISORROPIA predicted pH.*

In Zheng et al. (2021), the goal is to account for non-idealities in the thermodynamic calculation of pH based on ammonia partitioning. Note that Zheng et al. (2020) performed their calculations assuming ideal conditions. The non-ideality treatment of Zheng et al. (2021) is not actually thermodynamic model output, but is estimated based on three factors that are determined to impact non-ideality: temperature, RH, and the fraction of particulate anions contributed by nitrate nitrate. The estimation occurs then by interpolating from tables generated from thermodynamic models like ISORROPIA and E-AIM.

This correction was carried out on our data here. The ISORROPIA-generated thermodynamic lookup tables were used and the $pH_f$ (partitioning, non-ideal) were calculated and compared to the $pH_f$ (partitioning, ideal). While the non-ideality correction increases the correlation with ISORROPIA-calculated pH, we chose to not include it for three reasons: (1) the correction is based on interpolation from ISORROPIA outputs and attempts to mimic its output with more computational efficiency that running the full model; however, we do not need to mimic

ISORROPIA, as we have all of the necessary inputs and have already run it in full. We use the ammonia partitioning method as a completely independent thermodynamic calculation. That being said, we acknowledge that the method used here is based on an ideal assumption (the same assumption of ideality used throughout Zheng et al., Science, 2020). (2) The absolute impact on the pH calculated is not severe and does not impact the conclusions of the work (on average, a difference of ~0.5 pH units between partitioning ideal and partitioning non-ideal was observed). (3) Finally, we appreciate the Referee's comment and alerting us to this very recent manuscript. Given that Zheng et al. (2021) is currently under review, we feel that it would be premature to adopt their methodology before it has passed the peer-review process.  If substantial changes in their manuscript occur during revision (or if the manuscript is not accepted for publication), this would have serious implications for our conclusions if we adopt their methodology.

**2.) *Fig. 4, shows the relationship between aerosol pH and factors such as temperature, aerosol liquid water and total NH3. It seems that the influence of one factor on pH can also be affected by other factors. Is it possible to vary one factor with fixed other factors to investigate the influence of one factor?***

Pye *et al.* (2020) address this question in Table S3 of the accompanying supplemental information.  For five aerosol thermodynamic equilibrium models (E-AIM model III, AIOMFAC-GLE, MOSAIC, ISORROPIA II, and EQUISOLV II, the authors make predictions of molality-based pH and related properties for the water + $(NH_4)_2SO_4$ + $H_2SO_4$ + $NH_3$ system at 298.15 K.  For the case of their 'moderately acidic' water-free input system, increasing RH from 40% to 99% was demonstrated to increase ISORROPIA II-predicted aerosol pH from 1.84 to 4.47, with a corresponding change in the hydronium ion activity coefficient.  For the 'highly acidic' water-free composition input (50% $(NH_4)_2SO_4$ by mass, with less $NH_3$), similar change in RH resulted in aerosol pH increasing from -0.15 to 2.54.

While this experiment was performed with idealized systems, as described in the main body of their manuscript, we would expect similar findings with our observational results, *e.g.* increasing RH would increase aerosol pH at fixed concentration, and increasing the acidic components of the aerosol would decrease aerosol pH for fixed RH. However, doing so for our observations may not have much meaning, as ambient T, RH, and aerosol composition are fixed inputs at each time point for model inputs to predict aerosol pH.

Additionally, Tao and Murphy (2021), and Zhou et al. (2021), demonstrated the effects of individual drivers of aerosol pH on both diurnal and monthly/seasonal time frames utilizing ambient data by systematically decomposing the pH calculation. In most cases, temperature was the single largest driver, followed by $NH_3$ concentration, ambient RH, and aggregate particle properties in decreasing order of effect. Tao and Murphy (2021) found that temperature was the single largest driver of pH, followed by (in decreasing order of effect) the $NH_3$ effect (opposite the T effect), RH, and particle properties in Toronto, Canada. Likewise Zhou et al. (2021) found that temperature was the primary driver, both across seasons and day/night cycles, followed (generally) by the $NH_x$ effect, then RH and occasionally $SO_4^{2-}$, and during certain periods of the diurnal cycle, the NVC effect in decreasing order of effect.

**Minor Comments**

***1.) In Fig. S4, it would be better to use the same y scale when comparing pH values determined from different methods.***

We have combined the axes of the plots, ensuring the scales on the figure are identical for better comparison of the pH values.

***2.) Line 228-240: "The result shown in Fig. 4b is somewhat surprising because NH3 partitioning was quite sensitive to ALWC (Fig. 5); the relatively invariant aerosol pH is unexpected given the increase in NH3 uptake in the presence of ALW." A discussion of this surprising result would be useful. (The following discussion in the manuscript about the dry deposition is not very relevant to this result).***

We have retained the discussion on dry deposition, as we feel it is valuable to the overall discussion of aerosol pH (though not necessarily to the point we intend to make), but we have added additional discussion related to the surprising result as requested by the reviewer. The following discussion has been added to elaborate on the point we intended to make:

"With increasing NH$_3$ uptake in the presence of, and simultaneous with, the increased ALWC, it would be anticipated that aerosol pH would become more basic both through the reaction of NH$_3$ to form NH$_4^+$ and due to the dilution effect of liquid water. However, the results of Figures 4 and 5 reveal that despite both the increase in NH$_3$ uptake and increase ALWC, aerosol pH remains relatively unchanged, with only a 0.5 pH unit change at the highest values of ALWC."

This change also contains additional clarification to address Reviewer 1 Major Comment #6 as described above.

3.) *Line 296-298: ISORROPIA didn't give good NH3 partitioning predictions in this study and the different chemical compositions of the coarse- and fine-mode particles were used to explain it. I think the explanation is reasonable, however, I was still wondering what result you will get with E-AIM calculations. If the E-AIM also fails to predict NH3 partitioning here, this explanation would be more solid since E-AIM also assumes an internally mixed aerosol distribution.*

Making comparisons of ISORROPIA and E-AIM calculations is outside the scope of this particular work, and would be repeating work performed in recent studies. Notably, Pye et al. (2020) comprehensively compared multiple aerosol thermodynamic equilibrium models for consistency. In polluted environments, such as Mexico City, aerosol pH predictions obtained from E-AIM, ISORROPIA, and the NH$_3$ partitioning approach were shown to be in good agreement. Under both the moderately and highly acidic conditions described in Pye et al. (2020), E-AIM and ISORROPIA predicted values never exceeded 1 pH unit difference, and then only at low (<50% RH) values; for values above 60% RH this difference was on the order of 0-0.5 pH units. Exact values are given in Table 6 of Pye et al. (2020).

The pH of coarse-mode particles is anticipated to be higher than for fine-mode particles owing to the enrichment of the coarse-mode with NVCs from dust and sea salt. Differences of up to 4 pH units have been shown between fine- and coarse-modes. As E-AIM and ISORROPIA are shown to be in acceptable agreement at the observed ambient RH ranges in the present study (RH > 60%), it should be expected that ISORROPIA would more accurately predict the aerosol pH

when the coarse mode is included from an internally-mixed standpoint as E-AIM lacks NVC constituents as inputs.

**Urban aerosol chemistry at a land-water transition site during summer – Part 2: Aerosol pH and liquid water content**

M. A. Battaglia, Jr.[1,a], N. Balasus[1], K. Ball[1], V. Caicedo[2], R. Delgado[2], A. G. Carlton[3] , and C. 
[revised manuscript text omitted]

---

## Author Response (AR2)

*Response to Reviewer Comments for "Urban aerosol chemistry at a land-water transition site during summer – Part 2: Aerosol pH and liquid water content" by Michael A. Battaglia Jr. et al., Atmos. Chem. Phys. Discuss., https://doi.org/10.5194/acp-2021-368-RC1, 2021"*

We thank the Referee for their additional comments. We have addressed each comment with the Referee comments in bold and our reply in plain text immediately below.

**In the revised manuscript, the authors provided additional details and explanation to the discussion, which are generally acceptable. However, I still have some concerns about their data presentation and interpretation.**

**The QA/QC information the authors provided is not based on their own data, but from another citation. Without the calculation results associated with their own data, it is very hard to judge the uncertainty of pH calculations associated with their ionic compounds measurement. The author seems to claim one-unit difference is sufficient small in the response to major comment 1. However, their pH fluctuation range is also around 1 unit. I still insist that the authors should perform separate QA/QC calculations to quantify the uncertainties on pH calculation.**

We understand the Referee's concern here. We agree with the Referee that quantifying the uncertainty of aerosol pH calculations from thermodynamic equilibrium models represents an important priority in this research field. However, at present, challenges in the direct measurement of aerosol pH limit the extent to which uncertainty can be quantified in the pH calculations because uncertainty does not just derive from uncertainty in the measured ionic components. Pye et al. (2020) discuss this point extensively (see Sections 4 and 7). It is why the thermodynamic calculations of aerosol pH in Figures 3 – 7 of Pye et al. (2020) do not have error bars. Doing so would require a direct measurement of aerosol pH to quantify the model error. This is not a challenge that is unique to our study: for example, Zheng et al. (Science, 2020) do not present their thermodynamic predictions of pH with uncertainties, either (see their Figures 4b, 4c, S5, S6, S13b, and S14). We are following the standard convention described by Pye et al. (2020), and references therein.

**Figure 4 &5: even though the authors provide some explanations in the caption, it is still not clear about their logic to group data points through such approach. How the bins were chosen and why process through such approach?**

We have defined bins that are approximately evenly spaced across the range of observed values (in this case, temperature, aerosol liquid water, and total $NH_3$ concentration) with sufficient bin numbers and spacing so that trends in the mean values are apparent. We agree with the Referee that there is some level of subjectivity in the choice of bins in Figures 4 and 5. That is why we updated the figures so that all individual observations are plotted along with the bin means. We feel that the individual observations demonstrate that our primary conclusions are not sensitive to small changes in the bin definitions. However, all data in Figures 4 and 5 are publicly available (see data availability statement) so one could test different bin boundaries, if desired.

**References**

Pye, H. O. T., Nenes, A., Alexander, B., Ault, A. P., Barth, M. C., Clegg, S. L., Collett Jr, J. L., Fahey, K. M., Hennigan, C. J., Herrmann, H., Kanakidou, M., Kelly, J. T., Ku, I. T., McNeill, V. F., Riemer, N., Schaefer, T., Shi, G., Tilgner, A., Walker, J. T., Wang, T., Weber, R., Xing, J., Zaveri, R. A., & Zuend, A. (2020). The acidity of atmospheric particles and clouds. *Atmos. Chem. Phys.*, *20*(8), 4809-4888. https://doi.org/10.5194/acp-20-4809-2020

Zheng, G., Su, H., Wang, S., Andreae, M. O., Pöschl, U., & Cheng, Y. (2020). Multiphase buffer theory explains contrasts in atmospheric aerosol acidity. *Science*, *369*(6509), 1374. https://doi.org/10.1126/science.aba3719